# Vasculopathy and Coagulopathy Associated with SARS-CoV-2 Infection

**DOI:** 10.3390/cells9071583

**Published:** 2020-06-30

**Authors:** Nazzarena Labò, Hidetaka Ohnuki, Giovanna Tosato

**Affiliations:** 1Viral Oncology Section, AIDS and Cancer Virus Program, Frederick National Laboratory for Cancer Research, Leidos Biochemical Research Inc., Frederick, MD 21702, USA; labon@mail.nih.gov; 2Laboratory of Cellular Oncology, Center for Cancer Research, National Cancer Institute, National Institutes of Health, Bethesda, MD 20892, USA; hidetaka.ohnuki@nih.gov

**Keywords:** COVID-19, SARS-CoV-2, ACE2, inflammatory cytokines, vasculopathy, vascular inflammation

## Abstract

The emergence of severe acute respiratory syndrome coronavirus 2 (SARS-CoV-2), the causative agent of coronavirus disease 2019 (COVID-19), has resulted in >500,000 deaths worldwide, including >125,000 deaths in the U.S. since its emergence in late December 2019 and June 2020. Neither curative anti-viral drugs nor a protective vaccine is currently available for the treatment and prevention of COVID-19. Recently, new clinical syndromes associated with coagulopathy and vasculopathy have emerged as a cause of sudden death and other serious clinical manifestations in younger patients infected with SARS-CoV-2 infection. Angiotensin converting enzyme 2 (ACE2), the receptor for SARS-CoV-2 and other coronaviruses, is a transmembrane protein expressed by lung alveolar epithelial cells, enterocytes, and vascular endothelial cells, whose physiologic role is to induce the maturation of angiotensin I to generate angiotensin 1-7, a peptide hormone that controls vasoconstriction and blood pressure. In this review, we provide the general context of the molecular and cellular mechanisms of SARS-CoV-2 infection with a focus on endothelial cells, describe the vasculopathy and coagulopathy syndromes in patients with SARS-CoV-2, and outline current understanding of the underlying mechanistic aspects.

## 1. Introduction

A new strain of coronavirus, severe acute respiratory syndrome-coronavirus 2 (SARS-CoV-2), was identified in December 2019 and found to cause a severe respiratory illness in humans, called coronavirus disease 2019 (COVID-19). This disease rapidly spread worldwide and was recognized as a pandemic by March 2020. Previously, two other coronaviruses, SARS-CoV, identified in 2002, and Middle East respiratory syndrome MERS-CoV, identified in 2012, were found to cause severe respiratory diseases, named SARS and MERS, respectively, in endemic areas. No human SARS cases have been detected since 2004 and human MERS cases have steadily decreased from 2016 to July 2019 [1].

The respiratory diseases SARS, MERS, and COVID-19 have similar manifestations of fever, cough, and shortness of breath. Pneumonia is a common complication. Severe cases can lead to acute respiratory distress syndrome, particularly in the elderly with underlying diseases, which include diabetes, cardiovascular disease, and cancer. Additional, less frequent, manifestations include gastrointestinal symptoms [2,3]. Although a proportion of people infected with these coronaviruses has remained asymptomatic, the mortality rate for patients with SARS is approximately 15% and with MERS is approximately 35%. The case fatality rate for COVID-19 has been variably estimated between < 1% to 15% and evolving [3,4,5]. No effective vaccine for SARS-CoV-2 is currently available. The antiviral drug remdesivir has received an emergency use authorization for the treatment of suspected or laboratory-confirmed COVID-19 in patients with severe disease. Other drugs, such as recombinant ACE2 (APN01) and an anti-viral antibody cocktail (REGN-COV) are currently tested in clinical trials. The clinical course of SARS and MERS is remarkably similar, apart from an overall greater severity of pulmonary manifestations in patients with MERS than in patients with SARS [2]. COVID-19, however, appears to include an evolving set of clinical manifestation not previously reported in SARS and MERS, including stroke and cardiomyopathies associated with coagulopathy and vasculopathy, which can cause sudden death and other serious morbidities. Here, we provide the general context of the molecular and cellular mechanisms of SARS-CoV-2 infection, describe the vasculopathy and coagulopathy syndromes in patients with COVID-19, and outline current understanding of the underlying mechanistic aspects. It should be appreciated that this is a rapidly evolving field and that much is unknown about the distinctive epidemiology, clinical manifestations, and pathogenesis of these COVID-19 associated syndromes.

## 2. Functions of the Angiotensin-Converting Enzyme 2 (ACE2)

### 2.1. ACE2, a Regulator of the Renin-Angiotensin System

As discussed in Section 3, ACE2 is the primary cell surface receptor by which SARS-CoV-2 binds and enters cells. A consequence of SARS-CoV-2 infection is that ACE2 function is disrupted and this disruption contributes to the pathogenesis to COVID-19. To place this aspect of COVID-19 in an appropriate context, we will first review the molecular biology of ACE2. Human ACE2 is a transmembrane carboxypeptidase comprising a heavily glycosylated N-terminal ectodomain containing the enzymatic active site, a hydrophobic transmembrane domain, and a short intracellular C-terminal tail [6,7]. ACE2 belongs to the family of angiotensin-converting enzymes, which are essential regulators of blood pressure, cardiac function, and fluid balance [6,7,8,9,10]. Angiotensin-converting enzyme (ACE) is also a member of this gene family [11,12,13]. ACE2 and ACE have different biochemical functions [14]. ACE2 converts angiotensin I into angiotensin 1-9, a peptide with nine amino acids without known biological function, which can be further converted by ACE into the shorter angiotensin peptide 1-7, which is a blood vessel dilator [7,15,16]. ACE can also convert angiotensin I into angiotensin II, a peptide with eight amino acids which is a potent blood vessel constrictor that increases blood pressure. In addition, ACE2 can directly process angiotensin II into angiotensin peptide 1-7 [15,17]. Thus, whereas ACE produces angiotensin II, which induces vessel constriction and increases blood pressure, ACE2 is a physiological inhibitor of angiotensin II, and acts to reduce vasoconstriction and other biological activities induced by angiotensin II, as described in greater detail in Section 2.2.

Most of the activity of ACE2 is modulated via regulation of its expression on the cell surface, where the extracellular enzyme (called an ectoenzyme) is exposed to circulating vasoactive peptides [18]. ACE2 can also undergo proteolytic shedding to generate a soluble, catalytically active form with no known physiologic function [15]. Soluble ACE2 is detected at low levels in normal plasma [7,19,20,21] and at higher levels in the circulation of patients with hypertension [22], heart failure [23], severe respiratory syndrome [24], type1 diabetes [25], and other conditions [26,27]. Soluble ACE2 is also abnormally high in the cerebral spinal fluid of hypertensive patients [28]. Mechanistically, the shedding of ACE2 is attributed to the enzymatic activity of several members of the “a disintegrin and metalloprotease” (ADAM) family, particularly ADAM17/TNFα-converting enzyme (TACE), which is known to cleave several membrane-anchored proteins [28]. The biologically active, cell membrane-bound ADAM17/TACE also plays critical roles in the release of biologically active cytokines, such as TNFα and TGFα; chemokines, such as Fractalkine/CXC3CL1; cell adhesion molecules, such as ICAM-1; and receptors, such as IL-6R and IL15R [29].

Serum ACE2 levels are sex-dependent, with higher levels in males compared to females [25,30,31], and are potentially influenced by the ACE2 gene location on the X chromosome, sex hormones, and genetic factors [20]. In addition, serum ACE2 levels do not directly reflect levels of tissue-bound ACE2; however, some studies have shown an inverse relationship between circulating and membrane-bound ACE2 [32,33].

ACE2 mRNA is expressed in many human tissues [6,34], but most predominantly in the small intestine, testis, kidney, adipose tissue, and heart [35]. ACE2 protein has been identified by immunohistochemistry in the lung, alveolar epithelial cells [36], the heart muscle and coronary endothelium [7,16,35], the enterocytes of the small intestine at the brush border [36,37], the epithelium of stomach, duodenum and rectum [38], the kidney tubular epithelium of the proximal tubules and intrarenal vessels [7,14,36,39] and the basal layer of the epidermis [36]. Importantly, ACE2 protein has been visualized in the endothelial cells of capillaries, the small and large arteries and veins of numerous organs and in the arterial smooth muscle cells, which is consistent with low-level ACE2 mRNA detection in many tissues [6,36,40].

### 2.2. Functions of ACE2 in Endothelial Cells

By indirectly and directly catalyzing the conversion of angiotensin I and angiotensin II to angiotensin 1-7, ACE2 antagonizes angiotensin II, attenuating the vasoconstrictive, pro-inflammatory, pro-apoptotic, pro-thrombotic, mitogenic, metabolic and other vascular effects of angiotensin II [41,42]. The pro-inflammatory, hypertensive and other cardiovascular complications of excess angiotensin II are mediated by activation of the angiotensin II type 1 receptor (AT1R) [42,43]. The vaso-protective functions of angiotensin 1-7 are mediated by activation of the G protein-coupled receptor Mas, which functionally operates as an antagonist of angiotensin II [44,45].

Loss and gain-of-function experiments have shown that ACE2, through angiotensin 1-7, reduces endothelial cell production of reactive oxygen species, vascular adhesion molecule-1 (VCAM-1) and chemokine monocyte chemoattractant protein-1 (MCP-1/CCL2), and attenuates monocyte adhesion to endothelial cells and endothelial cell maturation into vascular structures [46,47,48]. Consistent with these observations, endothelial cells isolated from the aortas of ACE2 knockout mice displayed increased expression of inflammatory cytokines, including TNFα, IL-6 and MCP-1; adhesion molecules, including VCAM-1 and JAM-A; and the metalloproteases MMP-2 and MMP9 [49]. Similarly, bone marrow-derived monocyte/macrophages from ACE2 knockout mice showed increased expression of VCAM-1 and TNFα compared to controls [49]. In addition, thoracic aortic segments from ACE2 knockout mice displayed reduced relaxation in response acetylcholine [47]. When transduced into pulmonary microvascular endothelial cells, ACE2, through angiotensin 1-7, has been shown to reduce LPS-induced endothelial cell death and secretion of the pro-inflammatory TNFα and IL1β cytokines by inhibiting the JNK and NFκB pathways [50].

Consistent with these results, recombinant human ACE2 protected primary human endothelial cells from death induced by angiotensin II [51], and stimulation of endogenous ACE2 in human coronary arterial endothelial cells inhibited NF-kB signaling and reduced TNFα activity [52]. Similarly, adenoviral transduction of ACE2 or administration of angiotensin 1-7 reduced reactive oxygen species (ROS) production in endothelial cells and rescued endothelial cell function in diabetic mice [53]. In a mouse model of bleomycin-induced acute lung injury, systemic treatment with recombinant human ACE2 reduced pulmonary levels of the pro-inflammatory cytokines IL-6, TNFα and CXCL1/NAP3 (neutrophil-activating protein-3) cytokines, attenuated VEGFA-induced vessel permeability in the lung, increased pro-survival BCL2 protein levels, and reduced lung cell death, which improved lung function [54,55,56]. Thus, ACE2 has a range of anti-hypertensive, anti-inflammatory and antioxidant effects that oppose those of angiotensin II in the vasculature.

Angiotensin II-induced experimental hypertension is accompanied by increased thrombosis in large arteries [57], arterioles [58] and other blood vessels [59,60]. Thrombosis is a major complication of hypertension, which can be explained by several factors, including alterations in platelets, hypertension-related stress of cell components of the vessel wall, coagulation and fibrinolysis that promote a pro-thrombotic state [61,62]. Angiotensin II has been implicated as a mediator of thrombosis associated with hypertension, in part because patients treated with ACE inhibitors and angiotensin II receptor blockers have shown a lower incidence of stroke and thrombotic events compared to patients treated with other anti-hypertensive drugs [63,64,65]. Additionally, ACE inhibitors and angiotensin II receptor blockers corrected platelet [62], coagulation [66], and fibrinolytic [67] defects. However, the mechanisms responsible for the angiotensin II-dependent activation of the coagulation cascade in large arteries and microvasculature are incompletely understood, having been variously linked to increased plasminogen activator-1 levels, activation of type1 (AT1), type 2 (AT2), type 4 (AT4) angiotensin II receptors, and signaling from receptors for endothelin-1 and bradykinin [58].

A direct role of ACE2 in reducing thrombosis is supported by experimental results in mice, which showed that XNT, a small molecule ACE2 activator [68], reduced platelet attachment to injured endothelia and the size of thrombi, and delayed complete vessel occlusion in mice [69]. In addition, in a model of pulmonary hypertension, angiotensin 1-7, the catalytic product of ACE2, reduced thrombus formation in hypertensive rats [70]. Recombinant human (rh)ACE2 has undergone initial human safety trials [71], but—to our knowledge—no antithrombotic activity has been reported.

### 2.3. Non-Vascular Functions of ACE2

Besides its role in the pathophysiology of the vascular system, ACE2 plays a critical role in the heart, kidneys, and gastrointestinal tract [72,73,74]. Consistent with the wide distribution of ACE2 protein in the heart, particularly in the coronary vessels and to a lower degree in cardiomyocytes as well as in coronary vessels, ACE2 has been shown to play a primary role in the generation of angiotensin 1-7 in isolated rat hearts [75]. ACE2 deficiency resulted is cardiac hypertrophy and defective cardiac contractility, which were reversed by ablation of ACE [16,76]. In addition, administration of angiotensin 1-7 in ACE2-deficient mice was shown to be cardioprotective in an experimental model of heart failure [77].

In the kidneys, which overall express higher levels of ACE2 compared to the heart [35], ACE2 is present in the proximal tubular epithelium of the nephrons, and in the endothelial and smooth muscle cells of large interlobular arteries [26,78]. The global depletion of ACE2 in mice resulted in increased kidney inflammation associated with enhanced TGFβ/SMAD2/3 and NFκB signaling [79], increased albuminuria and development of glomerulosclerosis [73]. When ACE2 knockout mice were crossed with diabetic Akita mice, glomerular damage was more advanced than in non-diabetic mice [80].

ACE2 is also broadly expressed on the luminal surface of intestinal enterocytes [36,37], where ACE2 associates with the neutral amino acid transporter B0AT1 and is required for B0AT1 localization at this site [81,82]. When challenged with the intestinal irritant dextran sodium sulphate, ACE2-deficient mice developed increased intestinal epithelial damage associated with infiltration of inflammatory cells, bleeding, crypt damage, and diarrhea compared to the wild-type control mice [74]. In this setting, ACE2-deficiency was linked to defective regulation of intestinal amino acids homeostasis [74].

In the hematopoietic system, ACE2-derived angiotensin 1-7 has been shown to improve the survival, CXCL12-induced migration and proliferation of circulating CD34^+^ hematopoietic stem/progenitor cells from diabetic patients [83], which may contribute to vascular regeneration [84]. In addition, bone marrow-derived macrophages from ACE2 knockout mice showed an increased pro-inflammatory response to LPS stimulation [49].

## 3. ACE2 Functions as a Coronavirus Receptor

### 3.1. ACE2 Is a Receptor for SARS-CoV-2 and other Coronaviruses

ACE2 has been identified as the primary cell surface receptor for SARS-CoV-2 [85,86,87,88] (Figure 1). Previously, ACE2 was identified as the cell surface entry receptor for the coronavirus SARS-CoV and several SARS-related coronaviruses [89,90,91,92,93,94,95]. The spike (S) protein of coronaviruses, including SARS-CoV-2 [85,87], is a critical determinant of virus binding to ACE2 and virus entry into the target cell [86,96]. Structurally, the S glycoprotein, a homotrimer protruding from the viral surface, is composed of two functional subunits: the S1 subunit, which contains the receptor-binding domain and is responsible for binding to the cell surface, and the viral membrane-anchored S2 subunit, which contains the fusion machinery and is responsible for fusion of the viral and cellular membranes [97]. Once bound to the host cell membrane, intact S is cleaved by host proteases and undergoes extensive irreversible conformational changes believed to activate the protein for membrane fusion [98]. The cellular transmembrane protease serine 2 (TMPRSS2), furin and other host proteases cleave the SARS-S proteins, exposing the S2 site [85,87,99]. SARS-CoV-2 possesses a furin cleavage site at the S1/S2 boundary that is not present in SARS CoV and has been proposed to expand the cell tropism of SARS-CoV-2 compared to SARS-CoV [98]. An inhibitor of TMPRSS2 can block SARS-CoV-2 entry into cells [85], as does recombinant human ACE2 [100]. TMPRSS2 is more broadly expressed in human tissues than ACE2 [88,101,102]. Mouse polyclonal antibodies raised against the SARS-CoV S protein can also block SARS-CoV-2 cell entry [98].

Structural work has focused on the characterization of the receptor-binding domain of SARS-CoV-2 and its conformation when bound to ACE2. The crystal structure of the receptor-binding domain of the SARS-CoV-2 S protein in complex with ACE2 indicated a more compact conformation compared to the SARS-CoV S protein in complex with ACE2. This difference was correlated with divergence from SARS-CoV in several residues in the SARS-CoV-2 receptor-binding domain that potentially stabilize the receptor binding domain-ACE2 interface [103,104]. However, affinity measurements have shown that SARS-CoV-2 and SARS-CoV S proteins engage human ACE2 with similar kinetic rate constants and only slightly different dissociation constants [98]. Cryo-electron microscopy structures of ACE2-B0 AT1, the full-length human ACE2 co-expressed with the B0 AT1 amino acid transporter (B0 AT1 uses ACE2 as a chaperone in the gastrointestinal epithelial cells [81]) showed that two S protein trimers can simultaneously bind to an ACE2 homodimer, indicating a potential regulatory role of B0 AT1 and other ACE2-binding proteins in SARS-CoV-2 infection [105].

Since ACE2 is a critical determinant of SARS-CoV-2 infection and disease, genetic studies have looked for ACE2 variants with the potential to reduce the association of between ACE2 and the S-proteins of SARS-CoV [96] and SARS-CoV-2 [106]. A systematic analysis of the 32 coding region variants of ACE2 that could potentially affect the amino acid sequence of ACE2 showed no mutations affecting the residues of human ACE2 that are important for binding the S protein in coronaviruses, thus making it unlikely that ACE2 genetic mutants exist that confer resistance to S-protein binding [106]. However, analysis of the distribution of 15 unique expression quantitative trait loci (eQTLs) from the Genotype Tissue Expression (GTEx) database showed frequency differences among Chinese, East Asian and European populations, suggesting that genotypes of ACE2 gene polymorphism may be associated with different expression levels of ACE2 [106].

### 3.2. ACE2 Internalization and Shedding by Coronaviruses

Entry of SARS-CoV-2 and SARS-CoV into cells is associated with ACE2 internalization alongside viral particles and reduction of cell surface ACE2. Since the enzymatic activity of ACE2 resides in the N-terminal extracellular domain where it is exposed to circulating vasoactive peptides, ACE2 internalization is coupled with a shift in the balance of cell-associated vasoactive mediators toward increased angiotensin II, triggering vasoconstriction, inflammation and a pro-coagulant status. Once internalized, ACE2 can undergo angiotensin II-mediated ubiquitination and degradation [107]. Besides promoting ACE2 internalization, SARS-CoV promotes the enzymatic shedding of the ACE2 ectodomain [24,108], resulting both in the generation of a soluble form of ACE2 and an overall reduction of ACE2 content in the infected cells. Recombinant SARS-CoV S protein displayed the same ACE2-internalyzing and pro-shedding activity as the intact virus [108]. Importantly, ACE2 shedding has been documented in vivo at sites of SARS-CoV infection in the human airways [24]. By contrast, the S protein of HNL63 (NL63), a coronavirus that causes the common cold and also utilizes ACE2 as a receptor, caused neither ACE2 shedding nor production of TNFα, differences attributed to sequence divergence between their S proteins, which share only 21% identity [108,109]. As the S proteins of SARS-CoV-2 and SARS-CoV S share 76% amino acid identity overall and display a similar receptor binding module [104], it is likely that SARS-CoV-2 also induces ACE2 shedding (Figure 1).

The purified SARS-CoV spike protein can activate cellular TNFα-converting enzyme (TACE)/ADAM17 when it binds to cell surface ACE2; this prompts ACE2 shedding from the cell surface and release of a soluble form of the ACE2 ectodomain [108]. ACE2 shedding induced by TACE/ADAM17 is accompanied by the release of the biologically active pro-inflammatory cytokine TNFα [108] along with other pro-inflammatory cytokines, chemokines, and receptors [29] (Figure 1). Besides TACE/ADAM17, endotoxin and other stimuli can also induce ACE2 shedding [24]. Active TACE/ADAM17 facilitates SARS-CoV infection [108], but soluble ACE2 is dispensable for replication of SARS-CoV and NL63 [109].

Soluble ACE2 is detected at low levels in the circulation of healthy individuals [21,73] but at significantly higher levels in patients with cardiac disease [110] or with type 1 diabetes and vascular complications [25]. To our knowledge, no published information currently exists on soluble ACE2 levels in the circulation of patients infected with SARS-CoV or SARS CoV-2, although levels would be expected to be elevated based on ACE2 shedding induced by these coronaviruses. However, a study in twelve patients with COVID-19 reported significantly increased plasma levels of angiotensin II, which appeared to directly correlate with circulating SARS-CoV-2 RNA levels and degree of hypoxia [111].

Although no known physiologic function has been attributed to soluble ACE2 [15], experimental SARS-CoV infection was partially inhibited by a fusion protein of the ACE2 ectodomain [112]. Consistent with this, rhACE2 has been found to dose-dependently inhibit SARS-CoV-2 infection of target cells, when used at microgram/mL concentrations [100], providing a rational basis for considering soluble ACE2 treatment to interfere with SARS-CoV-2 infection [113]. This approach is now under investigation [114,115].

## 4. SARS-CoV-2 Infection of Endothelial Cells and Vascular Pathology

### 4.1. SARS-CoV-2 Infection of Endothelial Cells

SARS-CoV-2 can infect primary endothelial cells in culture. Using human endothelial cells derived from pluripotent stem cells and cultured as three-dimensional vascular organoids [116], Monteil et al. showed that SARS-CoV-2 could infect and replicate in the engineered vessels, and that rhACE2 added to culture inhibited this infection [100]. There is also some evidence that SARS-CoV-2 can infect endothelial cells in patients with severe cases of COVID-19 [117]. Electron microscopy detected coronavirus-like particles within capillary endothelial cells of the kidney glomerulus [117], but identification of SARS-CoV-2 particles by electron microscopy may be challenging [118]. These data suggested that the endothelium could be a secondary target of SARS-CoV-2 infection in humans and that the ubiquitous distribution of ACE2^+^ blood vessels may underlie the multi-organ pathology of COVID-19 [119]. A question is how SARS-CoV-2 ends up infecting endothelial cells.

Epidemiologic evidence indicates that the virus is transmitted among close contacts mostly through the respiratory route, presumably directly through infected droplet secretions or indirectly via fomites [119,120,121] (Figure 2). Indeed, bronchoalveolar-lavage samples from COVID-19 patients, the source of the original SARS-CoV-2 isolates and most subsequent isolates, infected primary cultures of human airway epithelial cell cultures [122]. The presence of SARS-CoV-2 RNA in nasopharyngeal or oropharyngeal swabs is currently used as an approved diagnostic tool to document infection. Sputum is also frequently positive for viral RNA [123]. SARS-CoV-2 detection appears to be more frequent than in patients with SARS and MERS [2,124]. The presence of SARS-CoV-2 in the lung of patients with COVID-19 has been documented by immunostaining [125], detection of viral RNA [123], and electron microscopy [126].

Another potential source of transmissible SARS-CoV-2 is fecal material [127] since SARS-CoV-2 RNA was detected in a proportion of anal swabs (4/16, 25%) [128] and stool specimens of COVID-19 patients (44/143, 29%) [123] (Figure 2). This is consistent with immunohistochemical detection of the viral nucleocapsid antigen in the epithelium of the stomach, duodenum, and rectum of a COVID-19 patient where ACE2 protein was also detected [38]. However, to our knowledge the presence of infectious virus in stools has not been documented. Transmissible SARS-CoV-2 could also derive from tear and conjunctival secretions, as viral RNA was detected in these samples from one of 30 COVID-19 patients with conjunctivitis [129]. Other studies have confirmed detection of viral RNA in the conjunctival sac [130] and ocular discharge [131]. Urine appears an unlikely potential source of transmission, since viral RNA was detected rarely in urine [123,132,133,134], but a case of infectious SARS-CoV-2 from urine has been documented [132].

There is evidence for SARS-CoV-2 dissemination to the circulation since viral RNA was detected in 6/41(15%) plasma specimens from patients with severe COVID-19 disease [3] (Figure 2). Other studies have reported lower (3/307, 1%) [123] or higher (11/50, 23%) [135], (6/16 %, 38%) [128], (6/10, 50%) [136] or (12/12, 100%) [111] frequencies of viral RNA detection in plasma of patients with COVID-19, likely reflecting differences in technique, sampling, and variations in disease severity.

Hematological spread of infectious SARS-CoV-2 to different tissues is supported by detection of high viral RNA titers in the liver, kidney, or heart, but not in the pharynx or saphenous vein, from 5/12 patients who had died from COVID-19 [136]. Also, SARS-CoV-2 was detected by immunohistochemistry in tissue-resident ACE^+^CD169^+^ macrophages of all (6/6) autopsy-derived spleens and lymph nodes of patients who had succumbed of COVID-19, but not controls [137]. Additionally, coronavirus-like particles were visualized by electron microscopy in pneumocytes of a patient who had succumbed of COVID-19 [126].

The full extent of organ distribution of SARS-CoV-2 is still unclear. However, a comprehensive analysis of SARS-CoV in tissues from patients who had died of SARS, showed that viral RNA could be detected in most tissues at variable levels, with few possible exceptions [138,139].

### 4.2. Vascular Pathology in Patients Infected with SARS-CoV-2

In addition to laboratory evidence that SARS-CoV-2 infects endothelial cells and that loss of endothelial ACE2 as a consequence of coronavirus infection confers a pro-inflammatory, pro-coagulant, and pro-apoptotic phenotype to endothelial cells, there is now emerging evidence of vascular pathology in most patients with severe COVID-19 (Figure 1). An autopsy series showed the presence of deep venous thrombosis involving the lower extremities bilaterally in 56% (7/12) of cases, associated with massive pulmonary embolism in four of these cases, which was listed as the cause of death [136]. Deep venous thrombosis at the extremities was accompanied by evidence of recent thrombosis in the prostatic venous plexus in 6/7 cases [136]. Post-mortem histopathologic analysis of lung tissue from 38 COVID-19 patients documented the presence of platelet-fibrin thrombi in lung arterioles of most (33/38) cases, congested capillaries (33/38) proximal to hemorrhagic alveoli often containing CD61^+^ megakaryocytes, and dense capillary foci presumably resulting from vessel intussusceptive sprouting (18/38) [126]. Similar histopathological observations came from post-mortem studies of lungs from 12 COVID-19 cases, describing the regular presence of microthrombi in small arteries, capillary congestion and areas of increased capillary density [136]. A comprehensive post-mortem analysis of multiple tissues (in 3 COVID-19 cases) described congestion and endotheliitis of small vessels with accumulation of mononuclear/lymphocytic cells around the capillary endothelium in the heart, small bowel, kidney, liver and lung; in one case, caspase 3-immunostaining revealed the presence of apoptotic bodies in endothelial cells lining the inner wall of inflamed blood vessels [117,126]. Another autopsy series on 67 COVID-19 cases described the presence of vascular microthrombi in multiple organs, particularly in the lungs (in 23/25 cases) where it was associated with capillary sprouting and capillary inflammation, and in the brain (in 6/20 cases) where it was associated with evidence of patchy acute infarcts [140]. As expected from vascular pathology that includes inflammation, congestion, thrombosis, hemorrhage and endothelial cell death, the surrounding hypoxic tissues had evidence of interstitial edema, damage, destruction, inflammation, fibrosis and vascular regeneration [117,126].

### 4.3. Cytokines and Coagulation Profile in COVID-19

A retrospective study of 21 patients identified significant elevations of plasma IL-6, IL-10, TNFα and IL-2 receptor in severe (*n* = 11) compared to moderate (*n* = 10) cases of COVID-19, whereas IL-1β and IL-8 levels were not significantly different [141]. Expanding on these observations, another study found that severely affected COVID-19 patients (*n* = 13) admitted to the intensive care unit (ICU) not only had higher plasma levels of IL-10 and TNFα compared to less severely ill patients not admitted to the ICU (*n* = 28) but also increased levels of IP-10, IL-2, IL-7, G-CSF, MCP1 and MIP1A [3]. These and other studies have concluded that the core pro-inflammatory cytokines TNFα, IL-1β, IL-6, G-CSF, GM-CSF as well as the chemokines MCP-1, IP-10 and MIP1α are elevated in patients with COVID-19, with higher levels in those patients who are critically ill compared with those with less serious illness [142]. When released rapidly, particularly from activated macrophages and in abnormally large amounts, IL-6, TNFα and other pro-inflammatory cytokines can cause severe clinical manifestations of high fevers, headache, low blood pressure, night sweats and multi-organ dysfunction referred to as “cytokine release syndrome”. This syndrome is present in severe cases of COVID-19 and is associated with a poor prognosis [141,143]; it was also found to be a principal cause of morbidity in patients infected with SARS-CoV and MERS-CoV [144]. Not unique to coronavirus infections, “cytokine release syndrome” is observed in other conditions, including patients with Castleman’s disease during flares [145] and in patients receiving chimeric antigen receptor (CAR) T cells [146].

Patients with COVID-19 typically display hemostatic abnormalities consistent with the presence of a severe coagulopathy that predisposes to thrombotic events and is directly correlated with disease severity [147,148,149]. In a cohort of 183 patients with COVID-19, the 21 patients who died (11.5%) differed from those who did not in having increased levels of D-dimer (a fibrin protein fragment; ~3.5-fold) and other fibrin degradation products (~1.9-fold) [147]. In a group of 22 consecutive COVID-19 patients admitted to the ICU, increased fibrinogen and D-dimer plasma levels, shorter clotting time and other coagulation abnormalities were significantly more frequent compared to healthy controls [149]. Consistent with the presence of a coagulopathy, deep vein thrombosis appears more frequently than expected in hospitalized COVID-19 patients, even with anti-coagulant prophylaxis [149,150]. Vascular platelet-fibrin clots are a common finding in COVID-19 autopsies [136]. Also, among patients who died with COVID-19, 71% met the International Society on Thrombosis and Haemostasis criteria for disseminated intravascular coagulation (DIC), compared to only 0.6% of the survivors [151].

IL-6, often abnormally elevated in the circulation of patients with severe COVID-19, is a pro-inflammatory cytokine produced by activated monocytes, macrophages, endothelial cells and other cells that has pleiotropic effects and plays a critical role in hemostasis [152,153,154]. Acting on hepatocytes, IL-6 promotes the synthesis of coagulation factors such as fibrinogen, tissue factor and factor VIII [153,155,156]; acting on the bone marrow, IL-6 stimulates megakaryocyte differentiation into platelets [157,158]; and acting on endothelial cells, IL-6 induces vascular permeability and other effects by stimulating VEGF secretion [159,160,161]. Unlike Kaposi’s sarcoma-associated herpesvirus encoded viral IL-6 [162] and p19 [163], which can directly activate gp130 signaling, cellular IL-6 requires binding to the IL-6R to stimulate gp130-Jak-STAT signaling [164]. Since soluble IL-6R is abnormally elevated in the plasma of COVID-19 patients, presumably secondary to cleavage from the cells surface by TACE/ADAM17 upon SARS-CoV-2 infection and the inflammatory response associated with COVID-19 [29], IL-6 can activate most cells, including endothelial cells, which do not express endogenous IL-6R but can be activated by the complex of soluble IL-6R and IL-6 [164,165].

An association between IL-6 and increased risk of vascular thrombosis and thromboembolism (Figure 2) is supported by experimental results and observational studies [166,167]. In COVID-19, adding to the blood clotting risk from increased IL-6, SARS-CoV-2 can infect, replicate and induce endothelial cell death, compromising the continuity of the luminal vascular surface; a perturbed endothelium can actively participate in pro-coagulant reactions [168]. Also, as a consequence of SARS-CoV-2 receptor binding and entry into any cells, cell surface ACE2 is reduced in association with increased angiotensin II, which confers a pro-inflammatory, pro-coagulant and pro-apoptotic phenotype to endothelial cells [50]. Thus, the mechanisms that sustain hemostatic abnormalities in COVID-19 are complex. Not surprisingly, management has been debated. An international collaborative of clinicians and investigators has recently reviewed general guidelines for the prevention and management of thrombotic events in patients with COVID-19 [151].

## 5. Vascular Manifestations of COVID-19

This section describes key epidemiological, clinical and laboratory characteristics of COVID-19 with an emphasis on relationships between these characteristics and the underlying physiopathology. Clinical and pathological manifestations of vascular involvement in SARS-CoV-2 are summarized in Table 1.

### 5.1. Demographics of COVID-19 Patients

In a meta-analysis combining the results of 147 studies on 20,662 Chinese patients [169], the mean patient age was 49 years and 53% of patients were male. Of the 8,028 patients for whom data was reported, 39% had one or more comorbidities, including hypertension (21%), diabetes (12%), cardiovascular disease (9%) and cerebrovascular diseases (6%). In a study of 4,103 COVID-19 patients in a single academic health system in NYC [170], the median age was 52 years and 51% were male; 15% had diabetes, 27% obesity and 30% cardiovascular disease. In the same study, comorbidities were found to be a powerful predictor of COVID-19-associated hospitalization. In a report from the Italian Institute of Health [171] 68% of 3032 decedents with COVID-19 had hypertension, 30% diabetes, 28% ischemic heart disease and 11% obesity; only 4% had no comorbidities. Hypertension, diabetes and obesity are often associated with ACE2/Angiotensin 2 deregulation. SARS-CoV-2 infection further aggravates this ACE2/Angiotensin 2 imbalance, which could suggest a specific role for these pre-existing conditions as risk factors for COVID-19 morbidity and mortality.

### 5.2. Pulmonary Disease

In the large meta-analysis from China discussed above, the most common clinical manifestations of COVID-19 were fever and cough, followed by fatigue, sputum production and shortness of breath [169]. The most common imaging and laboratory findings included abnormal chest CT scans (71 % ground-glass opacity and 30% consolidation), lymphopenia (48%), increased lactate dehydrogenase (42%), D-Dimer elevation (43%), and indicators of inflammation, including C-reactive protein and erythrocyte sedimentation rate. The most frequent complication was acute respiratory distress syndrome (ARDS) in 25% of cases. In a much smaller series of 24 patients with confirmed COVID-19 admitted to the ICU in the Seattle area [172], the most common symptoms were cough and shortness of breath; 50% of the patients had fever on admission. The mean age was 64 years, 63% were men; 58% had diabetes mellitus and 21% had chronic kidney disease. Chest radiography revealed bilateral pulmonary opacities in all patients tested (23/23). Moderate-to-severe ARDS was observed in 75% of these patients.

The median time to development of ARDS ranged between 8 to 12 days from onset of symptoms, with some COVID-19 patients progressing rapidly to ARDS [173]. A systematic search for pulmonary emboli in 26 COVID-19 patients on mechanical ventilation showed that they were detected in 23% (6/26) of patients, and venous thromboembolism (VTE) was detected in 69% (18/26) [174]. A lower incidence of VTE (25%) was reported in severe COVID-19 patients in another study [175].

Results of 12 consecutive complete autopsies of patients with COVID-19 concluded that massive pulmonary embolism was the cause of death in 4/12 cases and that VTE was present in 7/12 (58%) patients [136]. Another report on 11 randomly selected COVID-19 autopsies described macroscopic pulmonary thrombi in all cases with associated pulmonary infarcts in 9/11 autopsies [176]. Histologic evaluation noted the presence of multiple thrombi in small and mid-sized pulmonary arteries with evidence of parenchymal lung infarcts [176]. Similarly, thrombi/platelet aggregates were visualized microscopically in capillaries, arterioles and medium sized arteries of most COVID-19 lungs (21/23) [140]. These studies highlight the importance of thromboembolism in the pulmonary manifestations of COVID-19, either through embolization in the presence of VTE or through thrombotic events occurring in the pulmonary arterial circulation.

Vascular thrombosis in the lung is not a unique to COVID-19 patients, as it is also observed in patients who have been intubated for acute respiratory syndrome attributable to sepsis, toxic inhalation and other causes [177]. However, a post-mortem study comparing the lungs of 7 patients deceased from COVID-19 with the lungs of 7 patients deceased with ARDS attributable to influenza A and 10 uninfected control lungs [169] showed electron microscopy evidence that endothelial cells in COVID-19 lungs were infected with SARS-CoV-2 and displayed disrupted cell surface membranes and other signs of injury. The pulmonary capillary vessels of COVID-19 tissues contained microthrombi and evidence of intravascular (intussusceptive) neovascularization, which were more frequent than observed in the influenza cases. It is therefore likely that pulmonary vascular injury and thrombosis reflect a specific physiopathology of COVID-19.

### 5.3. Gastrointestinal Disease

Several studies have described the presence of abdominal symptoms, such as nausea, vomiting and diarrhea in a variable proportion (5–50%) of patients who also had respiratory symptoms of COVID-19 [178]. In a multicenter study of 318 patients with confirmed COVID-19 [179], 61% of patients reported at least one gastrointestinal symptom on presentation, most commonly anorexia (35%), diarrhea (34%) and nausea (26%). The patients with gastrointestinal symptoms tended to be overweight or obese and had other comorbid conditions. Gastrointestinal symptoms were the predominant complaint at presentation in 20% of patients and the only presenting symptoms of COVID-19 in 14%.

Since SARS-CoV-2 can infect the epithelial cells on the luminal side of the small intestine and the virus has been detected in stools and rectal swabs, gastrointestinal symptoms are potentially attributable to viral infection and replication in the intestine. It is currently unclear whether SARS-CoV-2 can be transmitted in rare instances via the fecal-oral route and can survive the low pH of the stomach and the detergent effects of the bile salts in the small intestine [180]. The virus could alternatively spread to the gut from the respiratory track through blood dissemination, but this is currently unconfirmed. Surgical resection specimens from four patients with COVID-19 with severe abdominal symptoms showed macroscopic evidence of bowel infarction or ischemia [181]. Histology of the specimens not only confirmed the presence of ischemic enteritis and patchy bowel necrosis but also showed the presence of thrombi and perivascular inflammation in the submucosal arterioles. The vascular pathology was considered causative of the bowel ischemia [181], raising the broader possibility that gastrointestinal symptoms in COVID-19 may have a vascular pathogenesis.

Liver involvement is also a potential contributor to COVID-19 pathogenesis. Histopathology of post-mortem wedge liver biopsies obtained in 48 COVID-19 cases showed evidence of intrahepatic vascular abnormalities in all patients, including partial or complete luminal thrombosis of portal and sinusoidal vessels, associated with portal vessel dilation, fibrosis and focal hepatic tissue hemorrhage and necrosis [182]. The presence of thrombi involving the portal venules was observed in 15/22 autopsies, occasionally associated with thrombi in some hepatic arteries and arterioles [140].

### 5.4. Neurological Disease

Oxley et al. reported on a small case series of five 33 to 50-year-old patients who presented with ischemic stroke during a two-week period (from March 23 to April 7, 2020) in a single center in New York City, and who were subsequently diagnosed with SARS-CoV-2 infection [182]. Although three of these patients had pre-existing risk factors for stroke, this incidence of stroke represented a seven-fold increase over previous comparable periods [182]. In a subsequent prospective series over the following month involving a group of New York City Hospitals [183], the age range of ten SARS-CoV-2-infected patients presenting with stroke had broadened to include patients 27 to 75 years old. Although none of these patients had pre-existing cerebrocardiovascular conditions, 8/10 had diabetes with or without hypertension and/or other predisposing comorbidities.

In a retrospective single-center study in Brescia, Italy, among neurological patients admitted in March, those diagnosed with COVID-19 (56 patients) were significantly more likely to have been admitted for stroke and transient ischemic attack (TIA) than those without the diagnosis of COVID-19 (117 patients), perhaps in part because the COVID-19 patients were older [184]. It is important to note, however, that a large study encompassing diverse areas of the United States reported a significant overall decrease in stroke admissions in March 2020 compared to 2018 and 2019 that had been preceded by a more modest increase in February. This decrease was potentially attributable to decreased admissions for mild strokes [185]. In addition, a retrospective analysis on 214 patients hospitalized with COVID-19 in Wuhan found that acute cerebrovascular disease was present in one of 126 patients with non-severe COVID-19 and in five of 88 patients with severe COVID-19 [186]. A post-mortem imaging study involving 19 patients who had died of COVID-19 disease showed subcortical bleeds in 2/19 cases and edematous changes in the cortex/subcortical regions in 1/19 cases. The presence of bleeds in these severe cases of COVID-19 suggested the occurrence of vascular pathology in these patients [187]. Furthermore, histological examination of 20 brains from COVID-19 autopsies revealed the presence of anoxic brain injury in two patients with the clinical diagnosis of stroke. In addition, 6/20 cases showed widespread microthrombi associated with evidence of patchy ischemic infarcts [140]. Diffuse microthrombi in the brain may underlie cognitive problems, headaches, and other symptoms in patients with COVID-19 even after the apparent resolution of the disease [140].

Overall, the current results are not sufficient to support an association between SARS-CoV-2 infection and stroke. Additional autopsy evidence will be instrumental to investigate possible causal relationship between SARS-CoV-2 infection and cerebrovascular pathology in COVID-19 patients, regardless of neurologic presentation.

### 5.5. Myocardial Disease

Several observational studies have reported a significant reduction in the number of hospital admissions for acute myocardial infarctions during the COVID-19 pandemic [188,189,190,191]. However, cardiovascular disease is prevalent in patients with COVID-19; 7% of all patients and 22 % of severe cases experienced myocardial injury [192]. Cases of cardiomyopathy have been reported in patients with COVID-19 [143,193,194,195], raising the possibility that myocarditis-related cardiac events may be increased in COVID-19 [143,196,197]. Consistent with cardiomyopathy, COVID-19 autopsy reports revealed the presence of a few mononuclear cell infiltrates in the myocardium, presumably recruited from the circulation through inflamed capillaries, but no obvious histological changes in the heart tissue [198]. In 3/25 cases, thrombi were detected in small epicardial vessels [140]. Cardiac biomarker studies, particularly elevated troponin levels, have suggested a high prevalence of cardiac injury in hospitalized COVID-19 patients [192,199,200]. In one study, atrial arrhythmias were commonly detected in patients with severe COVID-19 [201].

Due to the paucity of data, the incidence and pathogenesis of cardiomyopathy in COVID-19, particularly acute myocarditis, is currently unclear. However, cardiomyocytes are potential targets of SARS-CoV-2 infection as they express ACE2. Also, genetic and pharmacological studies in mice have shown that ACE2 regulates cardiac structure and function [202]; decreased ACE2 in SARS-CoV-2-infected cardiomyocytes could disrupt these activities, particularly impairing cardiac remodeling and increasing oxidative stress in cardiomyocytes [73]. In addition, genetic variants of ACE2 have been linked to left ventricular hypertrophy [73,203], raising the possibility that ACE2 may contribute to cardiovascular disease in COVID-19 beyond its role as a receptor for SARS-CoV-2.

### 5.6. Cutaneous Disease

In a prospective study involving 375 patients with confirmed or suspected COVID-19 during the peak of the epidemic in Spain, various skin manifestations were observed [204]. The most distinctive form, in 19% of patients, consisted of erythema-edema with some vesicles located on the extremities, reminiscent of chilblains. These manifestations affected the younger patients, appeared later in the course of mild forms of COVID-19 or in asymptomatic cases, and lasted about 10 days. Reports of “pseudo-chilblains” have also emerged from other studies of COVID-19 patients [205,206]. In one such case, the histology was similar to other chilblains of unknown cause, with edema of the papillary dermis and perivascular/peri-eccrine lymphocytic infiltrates but no evidence of endothelial cell damage [207]. The most common manifestation (47%) consisted of non-distinctive maculopapular lesions, which have many causes. A relatively rare manifestation (6%) in older and more severe COVID-19 patients consisted of livedoid/necrotic lesions that were reminiscent of occlusive vascular disease [204].

### 5.7. Kawasaki-Like Syndrome/Multisystem Inflammatory Syndrome in Children (MIS-C)

Kawasaki disease (KD) is a rare inflammatory disease of medium size arteries throughout the body that affects children usually under 5 years old, particularly children of Asian or Pacific Islander descent. The cause or causes of KD are unknown. The symptoms include fever, red yes, skin rash, swollen lymph nodes, abdominal pain, vomiting and diarrhea. Although most children recover without sequelae, KD is a leading cause of acquired heart disease in children as it may be complicated by inflammation of the coronary arteries and the heart muscle. SARS-CoV-2 infection can present with a syndrome resembling KD. This manifestation came to light when 10 cases of KD-like disease were diagnosed between March and April 2020 at the peak of the COVID-19 epidemic in Bergamo, Italy. The number of cases represented a significant increase compared with 19 cases of KD diagnosed in the previous five years at the same center [208]. SARS-CoV-2 serology (IgG or IgM antibodies) was positive in 8/10 patients; in 2/8 serology-confirmed cases, viral RNA was also detected in nasal swabs. The recent 10-case group differed from the 19-case historical group in mean patient age (7.5 versus 3 years, respectively), cardiac involvement (6/10 versus 2/19), the presence of “Kawasaki disease shock syndrome” (5/10 versus 0/19), and evidence of “macrophage activation syndrome” (5/10 versus 0/19). Confirming these results, a second study reported on 17 children (median age 7.5 years) admitted with a syndrome resembling KD in Paris during an 11 days period, which represented a significant increase over the previous year [209]. Fourteen of the patients had serological evidence of SARS-CoV-2 infection and seven also had viral RNA in nasopharyngeal swabs. Eleven/17 patients presented with “Kawasaki disease shock syndrome” requiring intensive care support, and 12/17 had myocarditis. All children recovered. At the onset, all had gastrointestinal symptoms and high levels of inflammatory markers. Given the apparent increase in the number of cases of KD-like syndrome in New York City and in the United Kingdom, advisories have been issued by the US Centers for Diseases Control and Prevention and the Royal College of Paediatrics and Child Health. The condition has been then named Multisystem Inflammatory Syndrome in Children (MIS-C).

## 6. Insights into Factors that Precipitate Vascular Pathology in COVID-19

Autopsies of patients who have succumbed from COVID-19 provide strong evidence that the vasculature in compromised in severe forms of the disease, as discussed in Section 4.2. Most commonly in the lungs but also in other sites distant from the lungs, including the small bowel, brain, kidney, heart, liver and skin, the vessels display a combination of thrombosis, hemorrhage, edema, inflammation, endothelial cell death and intussusceptive vascular sprouting. This pathology of COVID-19 affects arterial and venous blood vessels of various sizes and capillary beds but appears not to similarly involve all tissues. The reason for this selectivity is unknown but may relate to the transcriptomic diversity of endothelial cells in different organs and tissues [210]. It is possible that endothelial cells differ in their expression of critical determinants of SARS-CoV-2 infectivity, such as ACE2 and TMPRSS2 (Section 3.1 and Section 3.2), and other factors, such as responsiveness to inflammatory cytokines and ability to regulate the coagulation cascade (Section 4.3).

An analysis of vascular pathology in COVID-19 must consider that SARS-CoV-2 can infect and replicate in endothelial cells, as discussed in Section 4.1. Both endothelial cell infection and death have been documented in tissues of COVID-19 patients. An important and unresolved question is how frequently SARS-CoV-2 infects endothelial cells in patients and to which extent endothelial cell infection is responsible for the vascular pathology. Related to the direct contribution of viral infection, we do not know how SARS-CoV-2 reaches endothelial cells. Vascular pathology of COVID-19 is most prominent in the lungs (Section 5.2), which is not unexpected since in COVID-19 the respiratory epithelium is the site of primary SARS-CoV-2 replication with secondary destruction of the pulmonary parenchyma. The proximity between the infected respiratory epithelium releasing the virus and endothelial cells may favor local transmission. A similar situation may be operative in the small intestine (Section 5.3). However, hematogenous dissemination of the virus (discussed in Section 4.1) is more likely to explain the presumed presence of viral particles in the endothelial cells distant from the lungs, such as the renal glomerulus. Viral RNA is detected in the plasma of a sizeable proportion of patients and in many tissues, but we do not know whether infectious SARS-CoV-2 particles are present in the circulation. Consistent with this, the blood supply has been considered safe during the COVID-19 pandemic.

Despite these limitations, it should be noted that viral infection of endothelial cells could drive many aspects of the vascular pathology in COVID-19. First, SARS-CoV-2 can cause endothelial cell death through lytic replication, as discussed in Section 4.1. Second, to infect cells, SARS-CoV-2 binds to the cell surface ACE2 receptors, and in so doing compromises the vaso-protective functions of ACE2, an enzyme that attenuates the vasoconstrictive, pro-inflammatory, pro-apoptotic, pro-thrombotic and mitogenic effects of angiotensin II, as discussed in Section 4.2. The complex series of events surrounding viral infection is expected to locally activate components of the innate immunity network and compromise the integrity of the endothelial monolayer exposing the thrombogenic basement membrane; this in turn can result in the activation of the coagulation cascade (Section 4.3). Thus, by infecting endothelial cells SARS-CoV-2 would be expected to drive the characteristic vascular pathology of COVID-19.

The systemic increase of pro-inflammatory cytokines in many patients with severe COVID-19 is likely another major contributor to the vascular pathology in COVID-19 (Section 4.3). Monocyte/macrophage-lineage cells, a major cell source of many pro-inflammatory cytokines, are numerically increased and phenotypically activated in many tissues of patients with severe COVID-19 [140]. GM-CSF, often abnormally elevated in plasma of severe COVID-19 patients, likely is a main contributor to this pathology, as GM-CSF promotes hematopoietic stem/progenitor cell differentiation into monocytes in the bone marrow, their release into the circulation and their activation. Among other abnormally elevated cytokines in COVID-19, IL-6 stands out for its vascular pathogenic potential (Section 4.3). Acting on hepatocytes, IL-6 promotes the synthesis of fibrinogen, tissue factor and Factor VII. It also stimulates platelet production in the bone marrow. In addition, by promoting VEGF-A secretion from various cell types, IL-6 induces vascular hyper-permeability and sprouting angiogenesis. Thus, IL-6 can promote a pro-coagulant and pro-thrombotic vascular pathology, raising the possibility that an abnormally high level of IL-6 may contribute to the vascular phenotype in severe COVID-19. Yet, there are other diseases in which IL-6 is similarly elevated. In Castleman disease, IL-6 is elevated in conjunction with TNFα, IL-1β and IL-10 as in severe COVID-19; however, vascular manifestations are unusual in this condition [145]. Also, in patients receiving treatment with CAR-T cells who develop a cytokine release syndrome, DIC is observed, if infrequently, but vasculitis and other manifestations of vessel inflammation are not reported [211].

Despite current uncertainties regarding the relative contribution of SARS-CoV-2 infection and inflammatory cytokines to the development of vascular pathology in COVID-19, pre-existing vascular dysfunction attributable to hypertension-induced mechanical insults, metabolic-related stress and predisposition to thrombosis would be expected to magnify the vascular pathology in COVID-19. Consistent with this, epidemiological data indicate that old age, hypertension, diabetes mellitus and obesity increase the risk for severe COVID-19 associated with vascular manifestations (Section 5.1).

Immune dysfunction is not known to play a role in typical COVID-19, except for the severe lymphopenia in severe cases of COVID-19, which may relate to tissue recruitment of lymphocyte and redistribution away from blood, and perhaps a low or delayed type I interferon response [212]. However, a potential role of immune dysfunction is suggested in the context of MIS-C observed in children with evidence of prior COVID-19 infection. Although poorly understood, the inflammatory manifestations and distinctive topology of the affected vessels raise the question of whether the endothelial cells that reside in vascular beds affected by Kawasaki disease are distinctly programmed to respond to systemic inflammation and to locally propagate auto-immune responses.

## 7. Approaches to the Prevention and Management of Vascular Manifestations in COVID-19

The key contribution of vascular pathology to the severity of COVID-19, particularly in patients with defined pre-existing conditions, prompts the question whether vascular-targeted therapies could be useful to reduce morbidity and mortality of COVID-19. A number of clinical studies are now underway to test approaches (Table 1) to address the vascular component of the disease. These include anti-coagulant drugs to prevent thrombotic and thromboembolic disease; anti-inflammatory agents, such as steroids, interferon α2b, inhibitors of the JAK1/JAK2 kinases and of the Bruton tyrosine kinase; IL-6 or IL-6R inhibitors to neutralize IL-6 activity; anti-angiogenic agents targeting VEGF and angiopoietin 2; and renin-angiotensin-aldosterone system (RAAS) inhibitors. Some of these agents have already entered clinical practice, such as pharmacologic prophylaxis of venous thromboembolism for all hospitalized patients [151,213] and others have shown promise, especially steroids. Most of the others are actively being investigated in controlled trials. It is important to appreciate that many patients with pre-existing conditions receive a variety of drugs to control hypertension, coagulopathies and other pathologies. Concern has emerged that ACE inhibitors and angiotensin-receptor blockers (ARBs), which are commonly used to control hypertension, could increase the risk for SARS-CoV-2 infection because in some studies these drugs increased the expression of ACE2, the entry receptor for SARS-CoV-2, in the heart and kidney [214]. However, other studies have shown that treatment with ARBs may mitigate angiotensin II-mediated lung injury by blocking the AT1 receptors [215]. Similarly, the optimal management of COVID-19 patients with pre-existing thrombotic disease receiving anti-thrombotic drugs remains to be defined [151]. Results from ongoing investigations will hopefully resolve some of these controversies.

With the goal of developing effective strategies to prevent the occurrence of vascular disease or treat it once it has developed, in our view it would be very important to identify patients at high risk for serious vascular complications early in the course of their disease. We already know that risk factors for vascular complications include pre-existing vascular morbidities, evidence of high-level systemic inflammation, and perhaps a high viral load (Section 5.1). Reducing the viral load and consequently reduce the potential for endothelial cell infection could be of paramount importance. REGN-COV2, a two-antibody cocktail to the SARS-CoV-2 spike protein, which is required for cell infection, has entered clinical trials after having shown promising pre-clinical results in mice (NCT04315298). Immune serum from patients who have recovered from COVID-19 generally blocks SARS-CoV-2 infection of target cells and based on this property is currently FDA-approved for compassionate use in patients with severe COVID-19. However, the effectiveness of immune serum is currently unclear and is under investigation (NCT04321421). Recombinant human ACE2, the cell entry receptor of SARS-CoV-2, has been shown to competitively neutralize SARS-CoV-2 infection of cells, including endothelial cells in vitro. Currently, recombinant human ACE2 is being tested in initial trials (Table 2). In addition, the antiviral drug, remdesivir, which has received Emergency Use Authorization for the treatment of hospitalized patients with suspected or laboratory-confirmed, severe COVID-19 could also be beneficial to reduce replication and release of infectious particles (NCT04292730).

When present at abnormally high levels, IL-6 in conjunction with soluble IL-6R compromises endothelial cells and the vasculature in experimental conditions (Section 4.3), and IL-6 peak levels mark disease severity in COVID-19. Drugs that inhibit IL-6, IL-6R and JAK signaling (activated by IL-6 family members) are FDA-approved for the treatment of rheumatoid arthritis and other inflammatory conditions and are currently being evaluated in patients with COVID-19 (Table 2). There are complexities with the targeting of IL-6 in COVID-19. IL-6 is a multifunctional cytokine, which activates many cell types by distinct mechanisms and with different outcomes. A key factor is whether the IL-6R is soluble or cell-associated; and if cell-associated, whether the IL-6R is positioned in cis or trans relative to the transmembrane gp130 signaling mediator (Section 4.3). Endothelial cells usually do not express IL-6R but can respond to IL-6 when IL-6 is bound to either soluble IL-6R or to IL-6R present on the cell surface of another cell. This biochemical complexity is mirrored by the functional complexity and conflicting biological activities of IL-6 [216]. Besides its well-known role as a pro-inflammatory cytokine, IL-6 has been shown to have an anti-inflammatory role in myeloid cells by priming expression of IL-4RA and thereby changing their pro-inflammatory phenotype to an “alternative”, anti-inflammatory phenotype [217]. IL-6 also has an insulin-sensitizing and anti-inflammatory effect in response to physical exercise [218]. Because of these complexities, an indication for the use of drugs that neutralize IL-6 activity in COVID-19 will depend upon the results from controlled trials.

Appreciation of the central role of vascular pathology in COVID-19 prompts consideration of drugs that more specifically target the endothelium. IL-6 induces VEGF-A production, which promotes vascular hyper-permeability and vascular sprouting that are manifestations of the vasculopathy in COVID-19 (Section 4.3). VEGFA and VEGFR-neutralizing drugs are approved for use as anti-angiogenic agent in certain cancers and age-related macular degeneration. Clinical trials are underway to evaluate anti-VEGF drugs in COVID-19 (Table 2). The undesirable, if rare, adverse effects of VEGF/VEGFR targeting, which include hypertension and thromboembolic events, will need to be part of risk-benefit assessment in patients with COVID-19 who are predisposed to thrombotic events. Besides VEGF-A, another essential vascular regulator, angiopoietin 2, is currently being considered as a therapeutic target in COVID-19 (Table 2). The rationale for this approach is that circulating levels of angiopoietin-2 (ANG2) are abnormally elevated in severe COVID-19 patients [219]. This is consistent with the observation that inflammatory signals induce the rapid release of ANG2 from endothelial cells, particularly in the presence of TNFα, and that ANG2 inhibitors can reduce vessel inflammation [220].

Although currently speculative, identification of patients at high risk for developing vascular complications of COVID-19, early intervention to reduce viral load, and incorporation of vascular-specific drugs outline a rational approach to the prevention and treatment of vascular pathology in severe COVID-19 to be validated in controlled studies.

## 8. Concluding Remarks

In the span of just a few months, much has been learned about SARS-CoV-2, how it infects cells, and how it is so efficiently transmitted from person-to-person. Much less is clear about the spectrum of COVID-19 disease and the underlying mechanisms of the clinical manifestations attributable to SARS-CoV-2 infection, ranging from asymptomatic infection, disease localized to the upper respiratory tract, systemic disease, and fatal illness. One of the unifying links in many manifestations of SARS-CoV-2 infection is the systemic pro-inflammatory and pro-coagulant phenotype, associated with vascular thrombosis in arteries, veins, and capillaries and blood vessels inflammation. Pulmonary embolism, intestinal ischemia, vasculitis are serious manifestations of COVID-19 that are now coming increasingly into focus. This review represents an effort to explore the mechanistic aspects underlying vascular pathology in COVID-19. We highlight the essential contribution of SARS-CoV-2, which can infect and replicate in endothelial cells, in addition to other cells, and the predicted role of ACE2, the cell surface receptor of SARS-CoV-2, in creating an imbalance where increased angiotensin II promotes vasoconstriction and vascular inflammation. Systemic inflammation in COVID-19 patients is a participant effector that magnifies vascular pathology, as it includes the pro-coagulant mediators IL-6 plus IL-6R, which induce the vascular-permeability effector VEGFA.

## Figures and Tables

**Figure 1 cells-09-01583-f001:**
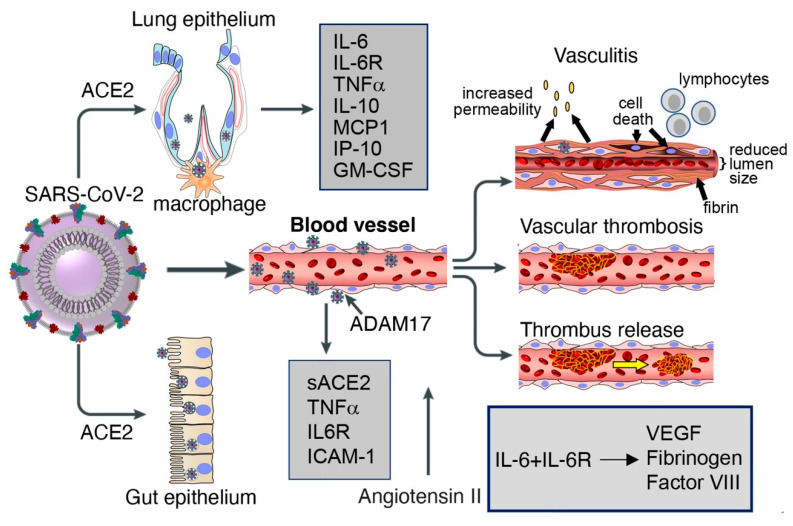
Lung and gut epithelia, macrophages, and vascular endothelia are infected by SARS-CoV-2 in COVID-19. Soluble (s) ACE2, inflammatory cytokines, cytokines receptors, chemokinesand other factors are released by virus binding and cell infection. Vascular pathology in COVID-19 includes vasculitis, associated with endothelial cell death, increased vascular permeability, recruitment of inflammatory lymphocytic cells, fibrin deposition and reduction of lumen size; vascular thrombosis; and vascular embolization.

**Figure 2 cells-09-01583-f002:**
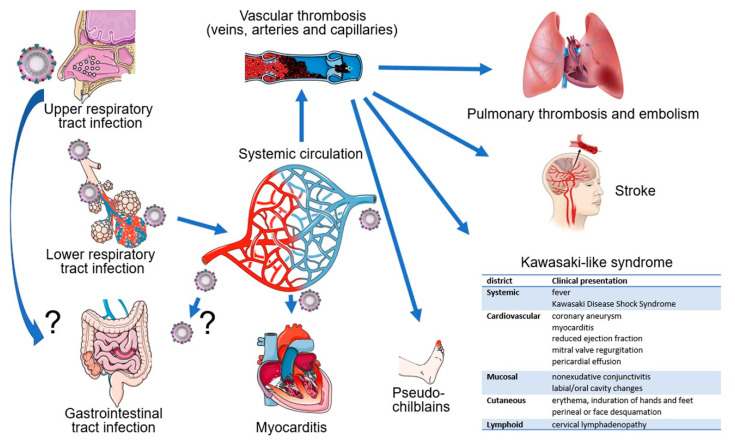
SARS-CoV-2 infects humans through the respiratory track by infecting and replicating in the epithelial cells that line the upper and lower respiratory track. In the lungs, the virus can cause COVID-19 pneumonia. SARS-CoV-2 can infect the intestinal epithelium in the small and large intestine; it is unclear whether this occurs through the oral route or dissemination from the respiratory track. SARS-CoV-2 can also infects myocardial cells. Hypercoagulation, vascular thrombosis and embolization are common in severe COVID-19 leading to pulmonary, brain and skin embolization. Kawasaki-like disease is linked epidemiologically to COVID-19.

**Table 1 cells-09-01583-t001:** Clinical and pathological vascular manifestations in SARS-CoV-2 infection.

System	Finding
Pulmonary	embolism
	microthrombi
	parenchymal infarcts
	intussusceptive neovascularization
Gastrointestinal	bowel ischemia/infarction
	thrombosis of portal/sinusoidal vessels occasional arterial thrombi
	hepatic hemorrhages/necrosis
Central nervous system	stroke, transient ischemic attack
	subcortical bleeds
	microthrombi, ischemic infarcts
Cardiovascular	cardiomyopathy
	venous thromboembolism
Skin	erythema pernio-like lesions, i.e., “pseudo chilblains”

**Table 2 cells-09-01583-t002:** Investigational treatments with potential impact on vascular manifestations of COVID-19.

Class	Drug/device	Trial Registration	Reference	
			Preprint	PMID
Anticoagulants	enoxaparin	NCT04406389 NCT04359277 NCT04427098 NCT04366960 NCT04377997 NCT04409834 NCT04400799 NCT04345848 NCT04401293 NCT04408235 NCT04373707	https://doi.org/10.1101/2020.03.28.20046144 https://doi.org/10.1101/2020.04.28.20082552 https://doi.org/10.1101/2020.05.30.20117929	32556875 32542212 32220112 32476080
	argatroban	NCT04406389	https://doi.org/10.1101/2020.05.30.20117929	32516429
Anti-inflammatory	corticosteroids	NCT04359511 NCT04329650 NCT04360876 NCT04273321 NCT04355247 NCT04273321 NCT04344288 NCT04348305 NCT04331054 NCT04416399 NCT04374071 NCT04355637 NCT04357860	https://doi.org/10.1101/2020.04.07.20056390 https://doi.org/10.1101/2020.03.06.20032342 https://doi.org/10.1101/2020.04.21.20066258 https://doi.org/10.1101/2020.06.15.20131607 https://doi.org/10.1101/2020.05.11.20097709 https://doi.org/10.1101/2020.06.17.20133579 https://doi.org/10.1101/2020.05.08.20094755 https://doi.org/10.1101/2020.04.17.20069773 https://doi.org/10.1101/2020.04.17.20064469 https://doi.org/10.1101/2020.05.22.20110544	32546811 32496422 32514354 32430057 32441786
Interferons	Interferon α2b	NCT04379518	https://doi.org/10.1101/2020.04.06.20042580	32483527
TNF-α inhibitors	infliximab	NCT04425538	https://doi.org/10.1101/2020.05.21.20108696 https://doi.org/10.1101/2020.04.30.20086090	32554621 32452979
IL-1 inhibitors	anakinra	NCT04362943 NCT04330638 NCT04366232 NCT04412291 NCT04362111 NCT04364009 NCT04339712 NCT04357366 NCT04324021 NCT02735707 NCT04341584	https://doi.org/10.1101/2020.06.16.20126714 https://doi.org/10.1101/2020.04.30.20086090	32501454 32438450 32437934 32437739 32422376 32411313 32376597
IL-6 inhibitors	tocilizumab	NCT04322773 NCT04345445 NCT04381936 NCT04409262	https://doi.org/10.1101/2020.05.01.20078360 https://doi.org/10.1101/2020.06.13.20130088 https://doi.org/10.1101/2020.05.29.20117358 https://doi.org/10.1101/2020.06.06.20122341 https://doi.org/10.1101/2020.06.08.20125245 https://doi.org/10.1101/2020.05.14.20099234 https://doi.org/10.1101/2020.06.05.20113738 https://doi.org/10.1101/2020.05.07.20094599	32557206 32553536 32515499 32482597
	siltuximab	NCT04329650	https://doi.org/10.1101/2020.04.01.20048561	
	olokizumab	NCT04380519		
	sarilumab	NCT04315298 NCT04386239 NCT04357808 NCT04341870 NCT04357860 NCT04359901 NCT04327388 NCT04324073	https://doi.org/10.1101/2020.05.14.20094144	32472703
JAK1/JAK2 inhibitors	ruxolitinib	NCT04366232 NCT04374149	-	32555296 32470486
	baricitinib	NCT04321993 NCT04399798 NCT04340232 NCT04401579 NCT04362943 NCT04421027 NCT04373044 NCT04390464 NCT04393051 NCT04358614 NCT04320277 NCT04399798 NCT03852537	-	32333918
Bruton tyrosine kinase Inhibitors	acalabrutinib	NCT04346199	-	32503877
	ibrutinib	NCT04375397	-	32302379
RAAS* drugs	recombinant human ACE2	NCT04287686 NCT04335136	-	-
	angiotensin II	NCT04408326	-	-
	ACE inhibitors	NCT04345406 NCT04322786	https://doi.org/10.1101/2020.04.28.20078071 https://doi.org/10.1101/2020.04.07.20056788 https://doi.org/10.1101/2020.06.11.20125849 https://doi.org/10.1101/2020.04.23.20076661 https://doi.org/10.1101/2020.03.31.20038935 https://doi.org/10.1101/2020.04.24.20077875 https://doi.org/10.1101/2020.05.19.20106856	32511678 32511473 32501480
	angiotensin II receptor blockers (ARBs)	NCT04337190 NCT04340557 NCT04408326	https://doi.org/10.1101/2020.04.28.20078071 https://doi.org/10.1101/2020.04.07.20056788 https://doi.org/10.1101/2020.06.11.20125849 https://doi.org/10.1101/2020.04.23.20076661 https://doi.org/10.1101/2020.03.31.20038935 https://doi.org/10.1101/2020.04.24.20077875 https://doi.org/10.1101/2020.05.19.20106856	32511678 32511473 32501480 -
	angiotensin peptide 1–7	NCT04375124 NCT04332666	https://doi.org/10.1101/2020.04.28.20078071 https://doi.org/10.1101/2020.04.07.20056788 https://doi.org/10.1101/2020.06.11.20125849 https://doi.org/10.1101/2020.04.23.20076661 https://doi.org/10.1101/2020.03.31.20038935 https://doi.org/10.1101/2020.04.24.20077875 https://doi.org/10.1101/2020.05.19.20106856	32511678 32511473 32501480 - -
Angiogenesis inhibitors	bevacizumab	NCT04305106 NCT04344782 NCT04275414	https://doi.org/10.1101/2020.04.28.20078071 https://doi.org/10.1101/2020.04.07.20056788 https://doi.org/10.1101/2020.06.11.20125849 https://doi.org/10.1101/2020.04.23.20076661 https://doi.org/10.1101/2020.03.31.20038935 https://doi.org/10.1101/2020.04.24.20077875 https://doi.org/10.1101/2020.05.19.20106856	32511678 32511473 32501480 -
	LY3127804	NCT04342897	- -	- -
Cytokine removal	CytoSorb	NCT04344080 NCT04391920 NCT04324528		
	Plasma exchange	NCT04374149 NCT04374539	-	32510799 32453903

* RAAS: renin-angiotensin-aldosterone system.

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
