# Peer review of "Vasculopathy and Coagulopathy Associated with SARS-CoV-2 Infection"

_cells, 2020, doi:10.3390/cells9071583_

Round 1

Reviewer 1 Report

Labo et al. have composed a comprehensive review addressing the vasculopathy and coagulopathy linked to SARS-CoV-2 infection.

I only have minor comments:

  1. line 38: mention that pneumonia is superimposed, since the virus per se does not induce pneumonia.
  2. line 44: at the time of review first data on the effectiveness of APN01 (rhACE2) that has been compassionately used in patients appeared, so that the authors may want to mitigate their statement that an effective treatment does not exist at all.
  3. line 96: it is the tubular epithelium of the proximal tubules.
  4. line 135-136: among the factors listed, I miss the endothelial dysfunction induced by hypertension-related stress of the respective cells as a risk factor for thrombosis.
  5. line 155: cardimoyocytes indeed produce ACE2, but the levels are negligible, compared to endothelia; the authors may want to mitigate this statement.
  6. line 161: epithelium of nephrons is not a good designation - please rephrase to proximal tubular epithelium
  7. line 256: the authors may check if the submitted paper of Penninger describing the effects of APN01 has not been already published.
  8. line 263: the authors may want to look at the serious criticism that has been risen respecting the reliability of the paper: Goldsmith CS, Miller SE, Martines RB, Bullock HA, Zaki SR. Electron microscopy of SARS-CoV-2: a challenging task. Lancet. 2020;395(10238):e99. doi:10.1016/S0140-6736(20)31188-0
  9. lines 300-303 and 423-428: authors may want take a note of one of the most systematic postmortem studies of COVID-19: Menter T, Haslbauer JD, Nienhold R, et al. Post-mortem examination of COVID19 patients reveals diffuse alveolar damage with severe capillary congestion and variegated findings of lungs and other organs suggesting vascular dysfunction [published online ahead of print, 2020 May 4]. Histopathology. 2020;10.1111/his.14134. doi:10.1111/his.14134
  10. line 316: though commented later (line 435), the authors may want to mention here that not only sprouting but also intussuceptive angiogenesis plays a role in COVID-19 associated vascular changes.
  11. lines 368-373: I am not so sure that IL6R elevation in COVID-19 patients may be explained only the action of ADAM17. Gene expression induction caused by the overall inflammatory response pattern of IL6R may also be a cause.
  12. line 455: there is no evidence of virus spread to the gut though blood dissemination now. It is suspected, as nicely described by the authors later, that there may be a spread to the splanchnic vascular endothelia causing ischemia via the mechanism of microthrombosis.
  13. lines 502-521: having performed many autopsies of COVDI-19 victims, I was not able to observe a single case of myocarditis snesu strictu. I am aware of the studies cited, but a careful examination of the accompanying images shows low evidence for the proper use of the designation myocarditis in this regard. In addition, larger autopsy studies, also those already cited by the authors, failed to uncover myocarditis. I would highly recommend attenuation of the discussion respecting myocarditis.

Author Response

Reviewer no. 1.

Labo et al. have composed a comprehensive review addressing the vasculopathy and coagulopathy linked to SARS-CoV-2 infection.

We thank the Reviewer for the time and effort in reviewing our manuscript and for the comments and suggestions for improvement, which we appreciate.

I only have minor comments:

  1. line 38: mention that pneumonia is superimposed, since the virus per se does not induce pneumonia.

We have modified the text to read:” Pneumonia is a common complication.” Line 75

  1. line 44: at the time of review first data on the effectiveness of APN01 (rhACE2) that has been compassionately used in patients appeared, so that the authors may want to mitigate their statement that an effective treatment does not exist at all.

We have modified the sentence to read: ”No effective vaccine for SARS-CoV-2 is currently available. The antiviral drug remdesivir has received an emergency use authorization for the treatment of suspected or laboratory-confirmed COVID-19 in patients with severe disease. Other drugs, such as recombinant ACE2 (APN01) and an anti-viral antibody cocktail (REGN-COV) are currently tested in clinal trials.” Lines 85-88

  1. line 96: it is the tubular epithelium of the proximal tubules.

Added: ”of the proximal tubules”. Line 141

  1. line 135-136: among the factors listed, I miss the endothelial dysfunction induced by hypertension-related stress of the respective cells as a risk factor for thrombosis.

Added: “hypertension-related stress of cell components of the vessel wall”; lines 180-181

  1. line 155: cardimoyocytes indeed produce ACE2, but the levels are negligible, compared to endothelia; the authors may want to mitigate this statement.

Changed to: “particularly in the coronary vessels and to a lower degree in cardiomyocytes”. Line 202

  1. line 161: epithelium of nephrons is not a good designation - please rephrase to proximal tubular epithelium

Modified to: ”proximal tubular epithelium”. Line 209

  1. line 256: the authors may check if the submitted paper of Penninger describing the effects of APN01 has not been already published.

Now published; reference updated. Ref. 100; line 1088.

  1. line 263: the authors may want to look at the serious criticism that has been risen respecting the reliability of the paper: Goldsmith CS, Miller SE, Martines RB, Bullock HA, Zaki SR. Electron microscopy of SARS-CoV-2: a challenging task. Lancet. 2020;395(10238):e99. doi:10.1016/S0140-6736(20)31188-0

We thank the referee for pointing to this publication. It is now mentioned in the text and quoted. Ref. 118; lines 268, 269

  1. lines 300-303 and 423-428: authors may want take a note of one of the most systematic postmortem studies of COVID-19: Menter T, Haslbauer JD, Nienhold R, et al. Post-mortem examination of COVID19 patients reveals diffuse alveolar damage with severe capillary congestion and variegated findings of lungs and other organs suggesting vascular dysfunction [published online ahead of print, 2020 May 4]. Histopathology. 2020;10.1111/his.14134. doi:10.1111/his.14134

We thank the referee for pointing to this publication, which is now presented and referenced. Ref. 139; line 313

  1. line 316: though commented later (line 435), the authors may want to mention here that not only sprouting but also intussuceptive angiogenesis plays a role in COVID-19 associated vascular changes.

Added, as suggested. Line 328

  1. lines 368-373: I am not so sure that IL6R elevation in COVID-19 patients may be explained only the action of ADAM17. Gene expression induction caused by the overall inflammatory response pattern of IL6R may also be a cause.

We have more fully addressed how soluble IL-6R can become abnormally elevated in plasma, including the role of inflammation. Soluble IL-6R elevations occur by one of two mechanisms, either by cleavage from the cell surface by active ADAM17 or increased expression of a variant splice form of the IL-6R (Scheller J et all Biochimica et Biophysica Acta 1813: 878, 2011). Inflammation can lead to ADAM17 activation and increased IL-6R shedding but the molecular basis is not clear (Scheller J. et al Trends in Immunol 32:8, 2011). Line 386; and Ref. 29

  1. line 455: there is no evidence of virus spread to the gut though blood dissemination now. It is suspected, as nicely described by the authors later, that there may be a spread to the splanchnic vascular endothelia causing ischemia via the mechanism of microthrombosis.

We clarified that virus spread to the intestine via blood dissemination has not been demonstrated. Line 173

  1. lines 502-521: having performed many autopsies of COVDI-19 victims, I was not able to observe a single case of myocarditis snesu strictu. I am aware of the studies cited, but a careful examination of the accompanying images shows low evidence for the proper use of the designation myocarditis in this regard. In addition, larger autopsy studies, also those already cited by the authors, failed to uncover myocarditis. I would highly recommend attenuation of the discussion respecting myocarditis.

In agreement with the Reviewer, we have further mitigated the potential occurrence of myocarditis in COVID-19. Lines 523-527

Reviewer 2 Report

This pleasant narrative review begins with a focus on the functions of ACE2, the main SARS-Cov-2 receptor. Next, evidences for infection of endothelial cells in COVID-19 are detailed. The article ends by a patchwork of most clinical/anatomical reports dealing with vascular manifestations occurring in COVID-19 patients.

NB : It would have perhaps been more appropriate : 1- to first describe more concisely (most salient information being gathered in a table) the vascular manifestations and thrombosis more frequently observed in COVID-19 patients than in patients infected by other coronaviruses; 2-then to detail the pro and con arguments for endothelial infection, and/or cytokine release, as most specific contributors to COVID-19 vasculopathies and coagulopathies; 3-last, to list the explanations and/or hypotheses already put forward to account for their more frequent occurrence in some organs, and/or subset of patients.

The strengths of this review are: 1-the extensive collection of clinical arguments for endothelial involvement in COVID-19; 2-the rather clear description of ACE2 biology and contribution of ADAM-17/TACE to ACE2 shedding.

Some weaknesses, however:

1-This review does not really try and explain why SARS-Cov-2 promotes more endothelitis and coagulopathy than other human coronaviruses, including SARS-Cov and MERS. It is just suggested page 6 that SARS-Cov-2 induces more ACE2 shedding than coronavirus causing common colds. However there is yet no evidence to support this hypothesis in the literature, which is also true for SARS-CoV and MERS. This is acknowledged by the authors page 6 (lines 245-6) « To our knowledge, no published information currently exists on soluble ACE2 levels in the circulation of patients infected with SARS-CoV or SARS-CoV-2 ».

Consequently, the statement in the conclusion (page 13, line 572) that SARS-Cov-2 « induces the shedding of ACE2 », although quite plausible, is still premature, and should be deleted.

Other limitations:

2-The sometimes conflicting findings of previous reports, which makes the text a bit confuse on several occasions: for instance: a- as to the expression of ACE2 mRNA or protein expression in various organs/tissues (lines 92-100) ; b- as to detection of SARS-Cov-2 in tissues (not found in spleen, lymph nodes, bone marrow, and heart (ref 136) (Page 8, lines 301-303), but found in the heart (Ref 134), or lymph nodes and spleen (Ref 135)).

3-The ‘shrinking’ of COVID-19 pathogenesis to interactions between SARS-Cov-2 S protein and ACE2 shedding. Because the shed form of ACE2 is catalytically active following SARS-Cov, and would counteract ACE-induced cellular signals, it is debatable whether ACE2 shedding by itself is a key event leading to tissue damage. In fact, the numerous interactions between viral proteins of SARS-Cov-2 and host intra-cellular pathways of innate immunity might be even more important. They could better account for the uneven distribution of the vasculopathy according to the organs, as well as the unusual intussuspective neo-vascularization more frequently observed in COVID-19 than in influenza (lines 438-9).

4-The lack of consideration of adaptive immune cells contribution to COVID-19 vasculopathy and coagulopathy, whereas T and B cells are mandatory in some animal models of virus-induced vasculopathies. The role of adaptive immune response as second trigger for vasculopathy might better fit with the delayed occurrence of vasculopathy and acute respiratory distress after the onset of first COVID-19 symptoms, than with the shedding of ACE2 receptor which should be maximal very early after SARS-Cov-2 infection.

5-The poor explanations given for the contrast between the paucity of vasculopathy in young people (excepted children with Kawasaki disease) and the frequency of vasculopathy/coagulopathy and cardiomyopathies in aged people, not previously reported in SARS and MERS. It would for instance be interesting to know whether the expression of ACE2, TACE and TMPRSS2, is affected by aging, either in the airway epithelium, endothelium, or other tissues.

Minor remarks

6-The statement (page 1, lines 44) that the case fatality rate for COVID-19 has been estimated between 3% and 15% should be seriously tempered, since hospital recruitment of the most severe cases accounts for this ratio. In the two reports on COVID-19 outbreak in cruise ships, where all (although rather old) passengers and crew have been tested by RT-PCR, less than 1% of infected people finally died: 1/128 with proven infection in South America (0.8%)(Ing AJ, et al. Thorax. 2020 May 27;thoraxjnl-2020-215091), and 2/696 (0.3%) in South-East Asia (Diamond Princess experience)(Kato H, et al. J Infect Chemother. 2020 May 13. doi: 10.1016/j.jiac.2020.05.005. Online ahead of print).

7-Page 2, lines 88-89. Possible explanations for the higher serum ACE2 levels in males could be provided, since: a/males are more susceptible to SARS-CoV-2 than females, with males 65% more likely to die from the infection than females; b/the ACE2 gene lays on the X-chromosome, which might lead to higher or at least equal ACE2 expression rate in females. The possibility that shedding of ACE2 and/or cleavage of the SARS-Cov-2 protein by the TMPRSS2 protease occur more easily in males could be discussed. The role of sex hormones on angiotensin production could also be briefly mentioned.

7-lines 341: IL-10 is not usually listed as a pro-inflammatory cytokine.

Author Response

Reviewer no. 2

This pleasant narrative review begins with a focus on the functions of ACE2, the main SARS-Cov-2 receptor. Next, evidences for infection of endothelial cells in COVID-19 are detailed. The article ends by a patchwork of most clinical/anatomical reports dealing with vascular manifestations occurring in COVID-19 patients.

NB : It would have perhaps been more appropriate : 1- to first describe more concisely (most salient information being gathered in a table) the vascular manifestations and thrombosis more frequently observed in COVID-19 patients than in patients infected by other coronaviruses; 2-then to detail the pro and con arguments for endothelial infection, and/or cytokine release, as most specific contributors to COVID-19 vasculopathies and coagulopathies; 3-last, to list the explanations and/or hypotheses already put forward to account for their more frequent occurrence in some organs, and/or subset of patients.

 The strengths of this review are: 1-the extensive collection of clinical arguments for endothelial involvement in COVID-19; 2-the rather clear description of ACE2 biology and contribution of ADAM-17/TACE to ACE2 shedding.

We thank the Reviewer for the time and effort spent in reviewing our manuscript, the laudatory comments and the insightful comments, which are all carefully considered in our revision.

Some weaknesses, however:

1-This review does not really try and explain why SARS-Cov-2 promotes more endothelitis and coagulopathy than other human coronaviruses, including SARS-Cov and MERS. It is just suggested page 6 that SARS-Cov-2 induces more ACE2 shedding than coronavirus causing common colds. However there is yet no evidence to support this hypothesis in the literature, which is also true for SARS-CoV and MERS. This is acknowledged by the authors page 6 (lines 245-6) « To our knowledge, no published information currently exists on soluble ACE2 levels in the circulation of patients infected with SARS-CoV or SARS-CoV-2 ».

Consequently, the statement in the conclusion (page 13, line 572) that SARS-Cov-2 « induces the shedding of ACE2 », although quite plausible, is still premature, and should be deleted.

As appropriately suggested by the Referee, we have modified the statement, which now reads:
“We highlight the essential contribution of SARS-CoV-2, which can infect and replicate in endothelial cells, in addition to other cells, and the predicted role of ACE2, the cell surface receptor of SARS-CoV-2, in creating an imbalance where increased angiotensin II promotes vasoconstriction and vascular inflammation.” Lines 734-738.

Other limitations:

2-The sometimes conflicting findings of previous reports, which makes the text a bit confuse on several occasions: for instance: a- as to the expression of ACE2 mRNA or protein expression in various organs/tissues (lines 92-100) ; b- as to detection of SARS-Cov-2 in tissues (not found in spleen, lymph nodes, bone marrow, and heart (ref 136) (Page 8, lines 301-303), but found in the heart (Ref 134), or lymph nodes and spleen (Ref 135)).

  1. We agree with the Referee that reports on ACE2 mRNA expression and protein levels are not always consistent in the literature; ACE2 mRNA has been reported virtually ubiquitously, whereas ACE2 protein has been detected only in certain tissues location. The reason for this apparent discrepancy is that the sensitivity of the two measurements is quite different (mRNA detection being much more sensitive) and the fact that ACE2 mRNA is generally expressed in the vasculature, which is present to a varying degree in all tissues and explains low levels detection of ACE2 mRNA in all tissues. To maintain accuracy, we report the results of the literature, even if at times there may be some discrepancy, which is explained by assay sensitivity. Lines 97-105
  2. The Referee is correct in pointing out that different studies have reported differences in the recovery of SARS-CoV-2 in different organs. This is expected to some extent when the patient populations studied differ in number, disease severity and the assay sensitivity differs. We have now changed the text to mitigate these potential differences. Lines 311-313

 3-The ‘shrinking’ of COVID-19 pathogenesis to interactions between SARS-Cov-2 S protein and ACE2 shedding. Because the shed form of ACE2 is catalytically active following SARS-Cov, and would counteract ACE-induced cellular signals, it is debatable whether ACE2 shedding by itself is a key event leading to tissue damage. In fact, the numerous interactions between viral proteins of SARS-Cov-2 and host intra-cellular pathways of innate immunity might be even more important. They could better account for the uneven distribution of the vasculopathy according to the organs, as well as the unusual intussuspective neo-vascularization more frequently observed in COVID-19 than in influenza (lines 438-9).

The Referee is correct in pointing out the complexity of the pathogenesis of vascular disease in COVID-19. This complexity is now extensively addressed in a new section (Section 6) titled: ”Insights into factors that precipitate vascular pathology in COVID-19”; lines 579-647. We agree with the Reviewer that the innate immunity is a likely contributor to vascular disease. This conclusion is strongly supported by endothelial cell release of mature and functionally active TNFa, IL-1b and other inflammatory mediators, which are important components of the innate immunity network triggered in response to DAMPs (also known as alarmins). SARS-CoV-2 infection accompanied by ACE2 internalization and ACE2 functional disruption is likely one of the initiating triggers for activation of innate immunity resulting in the release of inflammatory cytokines and other molecules. Please refer to previous sections 2.1 and 2.2 and the new section 6 describing these important pathogenetic aspects. We have clarified that soluble and catalytically ACE2 has no known function in vivo; only cell associated ACE2 has been found to physiologically catalyze the conversion of angiotensin 1 and angiotensin II to angiotensin 1-7, attenuating the biological activity of angiotensin II.

4-The lack of consideration of adaptive immune cells contribution to COVID-19 vasculopathy and coagulopathy, whereas T and B cells are mandatory in some animal models of virus-induced vasculopathies. The role of adaptive immune response as second trigger for vasculopathy might better fit with the delayed occurrence of vasculopathy and acute respiratory distress after the onset of first COVID-19 symptoms, than with the shedding of ACE2 receptor which should be maximal very early after SARS-Cov-2 infection.

The Referee is correct in pointing out the potential importance of defective adaptive immunity to the pathogenesis of many vasculopathies. We ourselves have worked on temporal arteritis where the contribution of adaptive immune de-regulation is clear (Espigol-Frigole G et al Science Signaling: 419, ra28, 2016). However, our analysis of the relevant literature suggests that a role for adaptive immune de-regulation in COVID-19 is currently unclear, except for the presence of lymphopenia generally explained by tissue recruitment of lymphocytes and redistribution away from blood, and perhaps an impaired type-I interferon response (Jamilloux , Y et al Autoimmun Rev 2020; new Ref. 212). Nonetheless, in the context of Kawasaki-type disease, the delayed appearance of the syndrome with respect to SARS-CoV-2 infection has suggested a potential link to autoimmunity. This is now specifically addressed; lines 640-647.

5-The poor explanations given for the contrast between the paucity of vasculopathy in young people (excepted children with Kawasaki disease) and the frequency of vasculopathy/coagulopathy and cardiomyopathies in aged people, not previously reported in SARS and MERS. It would for instance be interesting to know whether the expression of ACE2, TACE and TMPRSS2, is affected by aging, either in the airway epithelium, endothelium, or other tissues.

We fully agree with the Referee that it is very important that future research fully clarifies the biochemical mechanisms by which risk factors linked to the severity of COVID-19, including age, hypertension, diabetes and obesity, contribute to vascular pathology in this disease. This is now broadly discussed in the new section 6 (lines 579-647).  Analysis of expression of ACE2, TACE, TMPRSS2 in these high-risk individuals, as suggested by the Referee, and other important regulators of endothelial cell function, such as VEGF-A, VEGFR, neuropilin 1 angiopoietin-1, angiopoietin-2, TIE-1, TIE2, adhesion molecules would clearly provide an important set of data. It is important to note, however, that thrombosis is a well-known complication of hypertension, but the underlying mechanisms are still unclear, despite considerable effort, as discussed in our section 2.2.   

Minor remarks 

6-The statement (page 1, lines 44) that the case fatality rate for COVID-19 has been estimated between 3% and 15% should be seriously tempered, since hospital recruitment of the most severe cases accounts for this ratio. In the two reports on COVID-19 outbreak in cruise ships, where all (although rather old) passengers and crew have been tested by RT-PCR, less than 1% of infected people finally died: 1/128 with proven infection in South America (0.8%)(Ing AJ, et al. Thorax. 2020 May 27;thoraxjnl-2020-215091), and 2/696 (0.3%) in South-East Asia (Diamond Princess experience)(Kato H, et al. J Infect Chemother. 2020 May 13. doi: 10.1016/j.jiac.2020.05.005. Online ahead of print).

We thank the reviewer for pointing to this important study, which is now presented (line 40) and quoted (new Ref. 5).

 7-Page 2, lines 88-89. Possible explanations for the higher serum ACE2 levels in males could be provided, since: a/males are more susceptible to SARS-CoV-2 than females, with males 65% more likely to die from the infection than females; b/the ACE2 gene lays on the X-chromosome, which might lead to higher or at least equal ACE2 expression rate in females. The possibility that shedding of ACE2 and/or cleavage of the SARS-Cov-2 protein by the TMPRSS2 protease occur more easily in males could be discussed. The role of sex hormones on angiotensin production could also be briefly mentioned.

We now extend the text to mention additional explanations for higher ACE2 serum levels in males than in females. Lines 89-96.

 7-lines 341: IL-10 is not usually listed as a pro-inflammatory cytokine.

We agree with the Reviewer that IL-10 is usually considered an anti-inflammatory cytokine, often produced together with pro-inflammatory cytokines. Corrected (lines 352, 353).

Reviewer 3 Report

This is a very well written, informative and comprehensive review of SARS-CoV-2 infections from the perspective of ACE2 (and its deregulation) as the virus receptor. ACE2 expression is widespread in organs and tissues including endothelial cells. The authors focus on endothelial cells as a potential secondary target for SARS-CoV-2 infection and their role in the multiorgan infections and the associated vasculopathy and coagulopathy evident in severely affected COVID-19 patients. Comments; If infection of endothelial cells plays such a pivotal role in the SARS-CoV-2 widespread infection and severe disease, it is important to try to understand what factors precipitate viral infection of endothelial cells. Is it viremia or related to the patient comorbidity risk factors that already contribute to the ACE2/angiotensin II deregulation? Or do the virus induced effects on the host responses or immunopathology indirectly mediate adverse effects on the endothelial cells and their function? In this regard, a treatment section should be added at the end to address several aspects based on the extensive information presented in this review. 1. Many seriously ill patients are treated symptomatically with a myriad of drugs including corticosteroids, antibiotics, antivirals, pro-inflammatory cytokine inhibitors, etc. Is there any evidence that any of these might adversely (or beneficially) affect endothelial cell susceptibility or function or the ACE2/angiotensin II deregulation? 2. There is controversy about the impact of pre-existing treatments for several of the co-morbidities including the widespread use of ACE inhibitors or angiotensin receptor blocking drugs on COVID-19 disease or susceptibility. From the detailed perspective provided, it would be useful for the authors to comment on the potential impact of these drugs and also indicate any ongoing or future clinical trials needed to examine their effects or those of related possible treatments (recombinant human ACE2, etc) on COVID-19. It appears that few studies of COVID-19 patients monitor ACE2 or angiotensin II levels or activity. 3. Also from this extensive review, are there any additional pathways and relevant timeline for SARS-CoV-2 infection that could be targeted related to the secondary vasculopathy and coagulopathy syndromes that seem to be a major cause of COVID-19 fatalities? Additional comments: 4. In animal models of SARS-CoV-2 infections, most of which do not develop severe disease or die, do any show infection of endothelial cells or the associated vasculopathy and coagulopathy? 5. Lines 273-274 This sentence is unclear and contradicts numerous papers showing oropharyngeal or fecal shedding of SARS and MERS CoVs. 6. Line 361 COVID-19

Author Response

Reviewer no. 3.

This is a very well written, informative and comprehensive review of SARS-CoV-2 infections from the perspective of ACE2 (and its deregulation) as the virus receptor. ACE2 expression is widespread in organs and tissues including endothelial cells. The authors focus on endothelial cells as a potential secondary target for SARS-CoV-2 infection and their role in the multiorgan infections and the associated vasculopathy and coagulopathy evident in severely affected COVID-19 patients.

            We thank the Reviewer for time and effort in reviewing our manuscript and for his constructive comments. We have found the comments and suggestions extremely helpful and believe that the revisions we have made to address these comments have improved our review.

Comments;

  1. If infection of endothelial cells plays such a pivotal role in the SARS-CoV-2 widespread infection and severe disease, it is important to try to understand what factors precipitate viral infection of endothelial cells. Is it viremia or related to the patient comorbidity risk factors that already contribute to the ACE2/angiotensin II deregulation? Or do the virus induced effects on the host responses or immunopathology indirectly mediate adverse effects on the endothelial cells and their function?

The Reviewer raises a very important question, i.e. if vascular pathology is a critical determinant of COVID-19 disease severity -as reflected by histopathology reports in patients who have succumbed of the disease- how does it come about, how could be predicted and what treatment can prevent it. Although no clear answers are currently available, we now discuss this topic in detail and outline the information that is needed to answer currently open questions. This is presented in a new section (section no. 6) titled “Insights into factors that precipitate vascular pathology in COVID-19”. Lines 579-647.

  1. In this regard, a treatment section should be added at the end to address several aspects based on the extensive information presented in this review.
  2. Many seriously ill patients are treated symptomatically with a myriad of drugs including corticosteroids, antibiotics, antivirals, pro-inflammatory cytokine inhibitors, etc. Is there any evidence that any of these might adversely (or beneficially) affect endothelial cell susceptibility or function or the ACE2/angiotensin II deregulation?
  3. There is controversy about the impact of pre-existing treatments for several of the co-morbidities including the widespread use of ACE inhibitors or angiotensin receptor blocking drugs on COVID-19 disease or susceptibility. From the detailed perspective provided, it would be useful for the authors to comment on the potential impact of these drugs and also indicate any ongoing or future clinical trials needed to examine their effects or those of related possible treatments (recombinant human ACE2, etc) on COVID-19. It appears that few studies of COVID-19 patients monitor ACE2 or angiotensin II levels or activity.
  4. Also, from this extensive review, are there any additional pathways and relevant timeline for SARS-CoV-2 infection that could be targeted related to the secondary vasculopathy and coagulopathy syndromes that seem to be a major cause of COVID-19 fatalities? Additional comments.

The Reviewer suggested that we complement “the extensive information presented in this review” with a treatment section. We thank the Referee for this suggestion that we have followed. We have added a new section (section no. 7) titled “Approaches to the prevention and management of vascular manifestations in COVID-19”.Lines 648-685 and related new Table 2.

In this new section, we specifically address the important points raised by the Referee in his careful review. Briefly, we provide information on the drugs currently used in patients with COVID-19 either within or outside the context of clinical trials, and what is currently known about their potential efficacy (see also the new Table 2). As the Reviewer correctly noted, many drugs are currently used to treat patients with COVID-19, including drugs for pre-existing conditions. We specifically comment on the controversies surrounding the use of ACE inhibitors and angiotensin-receptor blockers (ARBs) for the treatment of hypertension in COVID-19 patients and how this controversy will be resolved (lines 661-665). In addition, we provide our perspective on selection of patients for treatment, the appropriate timing for initiation of treatment, choice of drugs and their potential impact for the treatment of COVID-19, with a focus on vascular disease manifestations. 

  1. Additional comments
  2. In animal models of SARS-CoV-2 infections, most of which do not develop severe disease or die, do any show infection of endothelial cells or the associated vasculopathy and coagulopathy?

To our knowledge, there is no animal model of SARS-CoV-2 infection, but there are animal models for the two related coronaviruses SARS-CoV and MERS-CoV. In a well-studied mouse model SARS-CoV, vascular pathology including perivascular “cuffing” composed of a mixed population of inflammatory cells was observed and this pathology was more severe in aged as opposed to young mice (Friedman M et al https://jvi.asm.org/content/jvi/86/2/884.full.pdf

  1. Lines 273-274 This sentence is unclear and contradicts numerous papers showing oropharyngeal or fecal shedding of SARS and MERS CoVs.

The sentence has been clarified. Lines 278-280.

  1. Line 361 COVID-19

We are unclear about this comment.

Round 2

Reviewer 2 Report

This article has been extensively modified and is quite suitable for publication now.